# Antibiotics and ECMO in the Adult Population—Persistent Challenges and Practical Guides

**DOI:** 10.3390/antibiotics11030338

**Published:** 2022-03-04

**Authors:** Francisco Gomez, Jesyree Veita, Krzysztof Laudanski

**Affiliations:** 1Department of Neurology, University of Missouri, Columbia, MO 65021, USA; fegyr7@umsystem.edu; 2Society for Healthcare Innovation, Philadelphia, PA 19146, USA; jassy.veita@shci.org; 3Department of Anesthesiology and Critical Care, University of Pennsylvania, Philadelphia, PA 19146, USA; 4Leonard Davis Institute for HealthCare Economics, University of Pennsylvania, Philadelphia, PA 19146, USA; 5Department of Neurology, University of Pennsylvania, Philadelphia, PA 19146, USA

**Keywords:** extracorporeal membrane oxygenation, ECMO, antibiotics, pharmacodynamics, pharmacokinetics, critical illness

## Abstract

Extracorporeal membrane oxygenation (ECMO) is an emerging treatment modality associated with a high frequency of antibiotic use. However, several covariables emerge during ECMO implementation, potentially jeopardizing the success of antimicrobial therapy. These variables include but are not limited to: the increased volume of distribution, altered clearance, and adsorption into circuit components, in addition to complex interactions of antibiotics in critical care illness. Furthermore, ECMO complicates the assessment of antibiotic effectiveness as fever, or other signs may not be easily detected, the immunogenicity of the circuit affects procalcitonin levels and other inflammatory markers while disrupting the immune system. We provided a review of pharmacokinetics and pharmacodynamics during ECMO, emphasizing practical application and review of patient-, illness-, and ECMO hardware-related factors.

## 1. Introduction

Extracorporeal membrane oxygenation (ECMO) has been increasingly employed in critical care, showing a reduction in 90-day mortality in ARDS vs. conventional care in mixed metanalysis [1]. However, other randomized control trials have failed to show benefits for ECMO deployment [2,3,4]. The interest in this emerging technology and widespread use seems to be slightly out of synchrony with the amount of supporting evidence [4,5,6]. In general, ECMO has found applications in several conditions characterized by unsustainable pathophysiology refractory to traditional therapies, including failures of pulmonary gas exchange or cardiac ability to maintain circulation [6,7,8,9]. 

The primary advantage of ECMO is to provide ventilatory or hemodynamic support in severely critically ill patients as a bridge to recovery in otherwise irrecoverable patients. The presumption is that stress related to ECMO implementation is less deleterious than mechanical ventilation or classical circulatory system support via implanted devices or medical therapy [6,10]. In that respect, ECMO provides “a bridge” to recovery by allowing sufficient time to surmount otherwise unsurvivable injury. A less common indication is to provide support during cardiopulmonary resuscitation or to preserve the viability of organs in donors [11,12]. 

A common indication for ECMO is acute respiratory distress syndrome (ARDS), most commonly from infectious etiopathogenesis [5,7,8]. In addition, sepsis is considered the indication for ECMO deployment in some cases [10,13]. Alternatively, patients undergoing ECMO may develop infectious complications that are byproducts of implementations [6,7,14]. The risk is relatively elevated considering the presence of invasive cannulation and emergence of immunosuppression secondary to critical care illness and considering the introduction of mechanical support devices [15,16,17].

ECMO introduces several variables into antibiotic pharmacokinetics and pharmacodynamics, which must be considered to maximize therapeutic benefit and minimize risks. Moreover, the effect of ECMO on said parameters may be further complicated by patient characteristics and concomitant illnesses or organ failure [18,19,20,21]. Therefore, adequate selection, management, and dosing of antibiotics and chemotherapeutics are challenging. Conversely, our review will clearly demonstrate that most of the data suggest that underdosing of antibiotics may lead to suboptimal outcomes. Alternatively, bactericidal antibiotics may attain a level typical for bacteriostatic levels rendering the adequate immune system critical for therapeutic success. 

The need for understanding how to optimize antibiotics effectiveness in ECMO-related situations is critical as the implementation and indications of the ECMO continues to progress, while the emergence of ECMO-derivative techniques such as a CO_2_-removal device, Impella, intra-aortic balloon counterpulsation, and cytokine scavengers add other variables to understanding distribution, activity and metabolism of antibiotics in these situations [6,7,22,23,24,25]. Given the increasing utilization of ECMO in the setting of systemic infection, an understanding of the interactions between said therapies and antibiotics is paramount to successful patient care. 

## 2. The ECMO Ins and Outs

ECMO is a relatively young modality that evolved from cardiopulmonary bypass [23]. In essence, ECMO can be considered as a protracted bypass and therapeutic takeover of pulmonary or cardiac function by mechanical devices. Driven by therapeutic goals, cannula configuration is applied to support the heart, lungs, or both. Venovenous VV-ECMO places both inflow and outflow cannulas in the venous system, allowing for gas exchange support in the absence of severe cardiac function impairment [23,26,27]. The ECMO circuit is integrated serially into the patient’s circulation in this configuration. Conversely, venoarterial ECMO (VA-ECMO) places the intake cannula in the venous system while the outflow is placed into an arterial vessel. This configuration supports lung and cardiac functions [14,23,27]. The circuit is placed in parallel to the heart, allowing for differential support of the cardiac function.

Cannulas provide an access port to the patient’s vascular system. They are single lumen and dual lumen [27,28]. To prevent kinking, they are made of metal coils embedded in protective shielding. Dual lumen cannulas need a precise placement but allow for higher mobility.

The ECMO system comprises several items in the circuit, with a pump and a membrane allowing for gas exchange as main components, connected via relatively high bore tubing [29] (Figure 1). The tubing is made of polyvinylchloride (PVC) with several coatings. Significant effort is taken to reduce a circuit-induced hypercoagulable state and immunogenicity via heparin or alternative coatings [30,31,32]. Transparency of the plastic tubing allows for visual inspection. Tubing pliability may lead to kinking and flow interruption, especially at 37 °C. A reinforced wire may be woven into the plastic to increase mechanical strength and to prevent kinking. The length of the tubing is dependent on circuit configuration, including additional elements (bridge, cytokine absorption devices, continuous renal replacement therapy bypass, access ports, and others) [33,34,35]. The length of the tubing is a compromise between ergonomics and patient mobility versus the overall need to minimize length [36]. The length of the tubing has several consequences. Apart from hemodynamics (i.e., shear stress, resistance to flow), tubing length determines the surface area coated by the biofilm, while length and diameter (3/8 inch) determine the fluid volume needed for priming as well as radiant heat loss.

The pump allows for high throughput, from the high bore intake cannula, through the oxygenator into the return cannula. There are two main types of pumps: roller and centrifugal [29]. The latter confers the advantage of minimized shear stress exerted upon erythrocytes [37,38]. The pump suctions venous blood from the patient, and a bladder may be introduced in front of the pump to prevent excessive negative pressure and venous collapse. The said pump produces the driving pressure necessary for blood to advance through the circuit and oxygenator while supporting perfusion pressure on the patient side. The pump design contributes to susceptibility of the circuity to kinking as the centrifugal pumps incur effluence with rising resistance, wherein the mechanical energy is lost as heat. In contrast, a roller pump, commonly found in CPB, will significantly increase pressure in a kinked circuit, leading to rupture. Safe pressure within the circuit is usually 300 mmHg, wherein 600 mmHg incurs the risk of rupture. 

The membrane oxygenator’s function is to provide a large surface area allowing for efficient gas exchange [39]. The effectiveness of the exchanger is measured as the amount of 75% saturated blood that can be further oxygenated to 95%. A gaseous mixture (usually oxygen and nitrogen) is injected into a gas exchanger. Carbon dioxide can be added for specific indications. The gas mixture is pumped through capillary tubing infused with blood, which flows counter to the gas [39]. Carbon dioxide exchange is quite efficient, while oxygen transfer is more limited due to the gases’ respective water solubility. The same principles govern this phenomenon as the gas exchange in the lungs. The reduction in the size of the oxygenators due to technological advances has resulted in fewer chances for blood pooling and thrombus formation.

Finally, a heat exchanger allows for precise and dynamic thermoregulation, and several in-line monitors and couplings allow for drug administration or system sampling [14,29]. There is also an increasing interest in providing additional support by introducing Impella, intra-aortic balloon counterpulsation, and bioabsorption devices, with significant implications for drug distribution [40].

In general, the evolution of the ECMO circuitry is reflected in a decreased form factor and lower immunogenicity of the hardware [41]. The former element has resulted in declining needs for volume fluid priming with direct effects on drug volume of distribution, including antibiotics. More compact form factors and lower immunogenicity limit the biofilm formation and drug absorption in the circuit. The design difference between leading manufacturers is usually related to user interface and design peculiarities with unclear, potentially negligible pharmacokinetics and pharmacodynamics.

### Infection and ECMO

Infection is the main driver for ECMO initiation, with meta-analysis of the CESAR and EOLIA trials finding ARDS to be the main indication for initiation of said therapy, with >60% being precipitated by pneumonia [1,6,7,8,13,23]. The risks factors for developing infection include more severally sick patients, ongoing immunosuppressive treatment targeting autoimmune diseases, prolonged cannulation, and VA ECMO [8,14]. In addition, critically ill patients develop a state of immunosuppression or anergy contributing to the infection’s risk [15,17]. At the same time, antibiotic effectiveness relies on the bactericidal effect instead of bacteriostatic or past-antibiotics effect in most critical care situations. 

Given the implantation of multiple invasive devices, ECMO itself confers risk for development of infections, including bloodstream infection at risk linearly related to the duration of therapy [14]. The prevalence of nosocomial infections in ECMO patients may range from 10–12% in registry data to 9–65% in single-center studies. Development of said infectious complications has been shown to increase morbidity and mortality, the latter by up to 38–63% [42]. In recent data, the most common sites of infection were respiratory at 56%, followed by bloodstream at 29%, and other sites, including urinary tract or soft tissues at 14% [43]. In more recent data, *Candida* sp. may have superseded other organisms [44]. Coagulase-negative staphylococci (15.9%), pseudomonas (10.5%), staphylococcus (9.4%), and *Enterococcus* (4%) are common pathogens [45]. Each hospital should have its profile for organism development. 

Currently, there is no recommendation for routine infection prophylaxis in ECMO patients [29], although some centers conduct routine blood cultures for surveillance [14]. Compounding the issue of cannula-related infection, cannulas cannot be easily, or in some cases feasibly, replaced [29]. Thus, appropriate care for cannulas and insertion sites is paramount to prevent iatrogenic infections. 

## 3. Antibiotics Therapy Principles

Antibiotic mechanisms of action can be classically divided into bacteriostatic, which inhibit bacterial replication while relying on the host’s immune system to clear the infection, and bactericidal, which lyse bacteria. These effects are highly dependent on free drug plasma concentrations and hence not only antibiotic selection. Dosing is also paramount to effective therapy. As bacteriostatic antibiotics rely on host mechanisms, immunosuppression or existence of a nidus or niche allowing unimpeded bacterial replication results in resumption of bacterial growth once the bacteriostatic compound reaches subtherapeutic levels. Thus, the application of said antibiotics in critical care is somewhat limited. However, many bactericidal antibiotics exercise bacteriostatic effects below their bactericidal concentration. Considering that ECMO and routine dosing of antibiotics depresses the concentration of antibiotics to bacteriostatic levels, thus maintaining the adequate function of the immune system, may be the next step in assessing the effectiveness of the antibiotic.

### 3.1. Pharmacokinetics and Pharmacodynamics of Antibiotics 

Antibiotic efficacy depends on several factors [46]. Most importantly, the concentration and the duration of exposure to antibiotics are critical. Pharmacodynamic properties of antibiotics will determine whether the majority of their bactericidal effects are concentration dependent, e.g., fluoroquinolones; time-dependent, e.g., beta-lactams; or a combination thereof, as the area under the curve dependent, such as glycopeptides [46,47].

The concentration of antibiotics is determined by the dose and the medium volume where the antibiotics are being diluted. Thus, the volume of distribution (Vd) is critical for determining antibiotic concentration [48]. The amount of the free drug is also determined by binding to circulating proteins or other molecules. The drug is then metabolized via several pathways involving liver, kidney, and other peripheral tissues [46]. Clearance (CL) is the fluid volume cleared from drug over a unit of time [46]. Most drugs undergo first-order kinetics, wherein a constant fraction of the drug is metabolized if the mechanism is not saturated. This is one of the critical determinants of the steady-state concentration of the drug [48,49].

Antibiotic concentrations can exert several actions depending on specific drug properties. The minimal bacteriostatic concentration (MBsC) relates to the minimum concentration that will inhibit bacterial replication in vitro and is utilized as a surrogate determinant of a specific antibiotic’s potency. Furthermore, bacteriostatic concentrations need to be sustained over time, as replication is impeded only under therapeutic concentrations. Consequently, antibiotic dosing must be frequent enough to prevent levels from dropping below MBC to maintain effectiveness. Conversely, increasing antibiotic concentrations diminishes returns despite bacteriostatic antibiotics exhibiting bactericidal activity at higher concentrations. However, the concentrations necessary for this effect to occur for these types of antibiotics are not feasible in this clinical setting. However, what is critical is the immune system’s performance to eradicate the bacteria. Bacteriostatic antibiotics retard bacteria growth, but eliminating the pathogens relies on immune system function.

The bactericidal effect refers to the direct killing of the pathogen. However, this effect depends on several factors. Minimal bactericidal concentration (MBC) is the level at which bacterial lysis begins to occur and is the determinant of a specific drug’s potency against the pathogen. As drug levels vary, a fall in concentration results in a predominant inhibitory, or bacteriostatic, action of the antibiotics, finally reaching a minimal inhibitory concentration (MIC) [46] (Figure 2). At this point, the bactericidal drug becomes bacteriostatic, and host defenses are necessary for the clearance of the microorganisms. 

Below MIC, drug actions do not necessarily cease. Several other antibacterial effects emerge, and the minimal concentration at which this effect occurs is called minimal antibacterial concentration (MAC). The post-antibiotic effect (PAE) refers to suppression of bacterial growth after a short pulse dose and has been previously described with several antibiotics and different bacterial strains [50,51,52] (Figure 2). Although MAC may guide antibiotic dosing, post-antibiotics effects are relatively short lived. In linezolid and ampicillin, the inhibition lasted between 1–3 h, depending on the type of bacteria treated [53,54]. For quinolones, the said effect may persist for up to 6 h [54]. Mechanisms are myriad and include inflicting sublethal damage, the persistence of antibiotics in periplasmic space, or efflux inhibition [55,56,57]. Post antibiotic leukocyte enhancement refers to increased bacterial susceptibility to immune system phagocytic activity [58]. Both bacteriostatic and bactericidal antibiotics can exercise this effect, but not all antibiotics can induce these effects [59,60,61,62,63]. The effect can be quite long for some aminoglycosides (tobramycin), allowing for one dose every 24 h [62]. Finally, MAC can trigger a reduction in pathogen virulence by modulating the immune response, altering chemotaxis adhesion, and decreasing pathogenic factor release [64,65,66,67]. These effects are sometimes grouped as post-antibiotics leukocyte enhancement (PALE) (Figure 2). The clinical effects of this phenomenon are unclear, as suppression of the immune system may occur concomitantly [68].

### 3.2. Limitations of Current Approaches to Monitoring Antibiotic Dosing

However, one must realize that antibiotic potency is measured in vitro under artificial conditions. The killing or bacteriostatic activity assessment is performed at pH of 7.2, in a protein-free, aerobic medium. Antibiotic activity is measured against 10^5^ of CFU during overnight exposure. These conditions diverge from physiological conditions in vivo. Notably, a plasma pH of 7.2 would signify severe acidosis and be considered an emergency. Catabolic processes during inflammation affect the circulating protein concentrations, while constant alterations in pH affect the electrostatic charge. Proteins abound in plasma, interacting with antibiotics in several ways, are highly variable in level and type, resulting from the ICU illness. Said factors are critical in dictating the amount of free antibiotic molecules that are critical for the antibacterial action as well as its potency.

The testing condition diverges substantially from the clinical reality of antibiotic dosing. A single dose of antibiotics is exceedingly rare in critical care situations. The bacterial load may be several-fold higher, and penicillin bactericidal properties are particularly sensitive to bacterial load. Most importantly, the in vitro test measures bacteria in the exponential growth phase, which is not necessarily the host’s phase. Measurement of antibiotic success is a change in physical properties of the growth medium, which may not be the best measurements of drug action or concentration translatable to the bedside. 

Conversely, measurements of antibiotics in serum in relation to antimicrobial activity may also be subjected to several biases. Poor penetration into bacterial nidus or sanctuary sites may necessitate increased dosages to achieve therapeutic concentrations within the target area. The ECMO circuit itself may offer a sanctuary for a pathogen to grow [69]. Furthermore, cellular antibiotic concentrations achieved are several-fold higher in some cases than those in plasma [70,71]. Certain biological compounds may inactivate other antibiotics. Measurements of sensitivity of bacteria rely on growth inhibition, but the concentration of antibiotics may change greatly depending on the fluid or compartment [70,72]. 

## 4. Critical Care Illness-Induced Changes in Antibiotics Levels

Several factors specific to ECMO further complicate the understanding of pharmacokinetics and pharmacodynamics of antibiotics in this setting. Some are related to critical care illness, while others are specific to the ECMO circuit itself.

Fluid resuscitation affects the volume of distribution, especially in the case of septic shock, where a large amount of fluid needs to be given to defend perfusion pressure despite venodilation and increase in vascular capacitance [73,74]. Endothelial activation secondary to an extracorporeal support circuit may promote capillary leakage increasing Vd [75]. Adding circuit volume and frequently pre-loading the patient to preserve the preload leads to a further increase in the volume of distribution (Vd) [36]. Liver and kidney failure can influence drug metabolism and excretion, and their function is highly dependent on ECMO performance, especially in VA ECMO [76]. Liver clearance is affected by blood stasis, which is highly dependent on the performance of the right ventricle [77]. Said performance may be affected by the emergence of cor pulmonale due to hypoxia, one of the primary reasons of ECMO implementation [6,7,8]. Fluid resuscitation can further exacerbate venous liver congestion [77,78]. The significant increase in fluid balance results in excessive mortality in ECMO [74]. Several factors mentioned above likely play a role. Secondarily sick patients may suffer from hypoalbuminemia, unpredictably affecting the level of free antibiotics [79]. Furthermore, the composition of the protein and the charge may be significantly different as seen in the nominal condition. 

## 5. Antibiotics in ECMO

The interaction of the antibiotics during ECMO is complex and most likely results in a suboptimal level of the antibiotics (Figure 3). In addition, concomitant immunosuppressive conditions further hamper the ability of the patients to recover fully.

### 5.1. Pharmacokinetics

Notably, since ECMO is an emergent treatment, large, randomized trials or even case series testing for pharmacodynamic or pharmacokinetic alterations concerning antibiotic microbial effectiveness in this population are lacking. Most of the data reported arise from observational trials.

Patients on ECMO may exhibit various and wide-ranging alterations in pharmacokinetics, some attributable to said treatment and others related to the critical illness itself [75,80]. Altered parameters noted ex vivo have included decreased half-lives and clearance and increased Vd. Some of these effects may be attributable to circuit sequestration [75,81]. For example, it has been well described that patients on ECMO may require higher doses of sedatives and analgesics, a phenomenon that carries over to several antibiotics. In addition, numerous studies in animals, neonates, and adults have shown significant variability and unpredictability in antibiotic pharmacodynamics during ECMO therapy [80,81,82,83].

Antibiotic strategies not accounting for these changes carry an increased chance of treatment failure, both instances of underdosing and supratherapeutic levels causing side effects, which have been reported [43]. In addition, suboptimal antibiotic dosing becomes dire in these patients due to the progression of the primary process, while selective pressure for the development of antibiotic resistance renders antibiotics less useful on the population level [82].

### 5.2. ECMO Specific Patient-Related Factors Affecting Antibiotics Distribution

The critical illness itself may incur fluid status dysregulation, thus an increase in the volume of distribution [80]. It has been noted previously that large variations in pharmacokinetics in critically ill patients occur between and even within the same patient [75]. Renal or hepatic impairment may decrease drug clearance and decrease pulmonary blood flow [44,82]. Setups producing no pulsatile flow may stimulate the renin–angiotensin–aldosterone axis, increasing fluid retention [44]. Additionally, lack of pulsatile flow decreases the glomerular filtration rate [81]. These patients’ conditions are dynamic and fluctuate rapidly.

### 5.3. Performance of the Immune System

Activation of the immune system may be altered in a way that is difficult to characterize at the current state of knowledge. This may significantly affect antibiotics’ MIC and MAB levels. In addition, some of the medications administered during ECMO may have additional antibacterial effects. For example, non-inflammatory nonsteroidal drugs alter the activity of Gram(+) bacteria and may enhance the antibiotic’s effect and modulate immune system activity [84,85,86]. In addition, proton pump inhibitors have additional antimicrobial activities, which are difficult to assess in terms of clinical efficiency [87]. 

### 5.4. ECMO Specific Hardware-Related Factors Affecting Antibiotics Distribution

#### 5.4.1. Circuit-Related Factors

Various circuit parameters may alter pharmacokinetics (Figure 3). These phenomena depend on drug properties, circuit type, roller, and biofilm formation [3,4]. The ECMO circuit comprises a large surface area that may sequester drugs, with circuit coatings and components themselves allowing for the adsorption of antimicrobials, thus reducing bioavailability [18]. This effect may be more pronounced in lipophilic drugs, although this effect may wane as binding sites saturate. This may also result in the circuit acting as a reservoir with subsequent redistribution into plasma [82,88]. Lipophilic drugs tend to be most readily sequestered in the circuit [80,82]. Meropenem is heavily sequestered (80%), most likely affecting its anti-bacterial potency [89,90,91]. Similar sequestration is seen for cefazolin, ampicillin, gentamycin, voriconazole, and vancomycin, but most of the studies were performed in vitro [89,91,92,93,94]. However, in the case of cefazolin, the in vivo study failed to demonstrate a lower level of drug [95]. Oxygenator seems to be particularly absorbent for some antibiotics, which is related to high surface area of the device and properties of membranes [96,97,98]. Silicone-constructed membranes have exhibited more drug residues than those composed of hollow fibers [44]. Other ECMO-dependent factors include priming fluid selection, which may incur less pronounced effects in adults than in neonates [44,75]. However, the effect of biofilm formation on the ability of the membrane to sequester antibiotics cannot be ascertained. These factors may be further complicated by concomitant cytokine absorption techniques or co-existing renal replacement therapies [22,34,99,100,101].

#### 5.4.2. Drug-Related Factors

Various properties of specific antibiotics directly influence ECMO effects on their pharmacodynamics. These include whether the antibiotic itself is lipophilic or hydrophilic, the tendency for protein binding, and the site of metabolic breakdown (Table 1) [82,102,103]. Furthermore, target MIC may vary by an agent or pathogen sensitivity.

## 6. Selected Antibiotics

Vancomycin is a hydrophilic glycopeptide antibiotic with bactericidal properties and low protein binding [43,47]. As clearance of this antibiotic is closely related to that of creatinine, it is usually dosed [47]. A wide variability for vancomycin Vd in ECMO patients has been noted previously [81]. An in vitro study suggested sequestration of vancomycin [94]. Analysis of retrospective data suggested no significant difference in drug concentration, Vd or clearance in ECMO vs. non-ECMO patients [104]. Vancomycin pharmacodynamics are largely unaffected by ECMO in several studies [103,105,106]. These results are not universal, as Park et al. demonstrated decreased levels in ECMO patients despite similar elimination rates, as seen in prior studies [106,107]. Wu et al. showed the opposite result in the affected clearance rate but showed unchanged pharmacokinetics parameters [108]. Differences in age or hardware use may account for these extremely heterogeneous conclusions. Current recommendations are: loading dose of 25–30 mg/kg followed by 15–20 mg/kg q81–2h dosage, as guided by therapeutic monitoring [43]. Another proposed regimen specifically for methicillin-resistant staphylococcus aureus recommended 400 mg q8h for MIC ≤ 0.5 µg/mL, or 600 mg q8h if the MIC was ≤1 µg/mL [103]. 

Amikacin is a hydrophilic aminoglycoside with bactericidal and post-antibiotic inhibition effects [47], with a low degree of protein binding [43]. While it has been posited the effects of ECMO on amikacin pharmacodynamics may be minimal, critically ill patients exhibit an increased volume of distribution. Studies involving gentamicin, another aminoglycoside, have noted a slight increase to a 1.5-fold increase in Vd for this population [81]. This said phenomenon exhibits a linear relationship in disease severity. One prospective observational study compared nine ECMO patients vs. 30 undergoing RRT vs. 50 with preserved renal function, wherein pre- and post-dosing amikacin concentrations were measured within 96 h. An increased volume of distribution and decreased clearance was observed in the ECMO group [109]. A similar study included 46 ECMO patients and controls and measured peak levels at 30 min after dosing and at 24 h, finding no significant differences in either measurement between said groups. An amikacin loading dose of 45–30 mg/kg is recommended [29,43], and given the narrow therapeutic window for aminoglycosides, routine therapeutic monitoring and further dosing are recommended as guided by achieved levels [43]. Given its narrow therapeutic window, the latter is paramount [47].

Meropenem is a carbapenem antibiotic, with effects similar to that of beta-lactams, exhibiting both bactericidal and post-antibiotic inhibitory effects [47,60]. Protein binding is low [47]. Several studies have demonstrated significant sequestration of the drug by circuit in vitro [110]. While increases in both volumes of distribution and clearance are likely, several trials failed to show a significant difference in pharmacodynamics in ECMO patients [47,75,80]. One study comparing 26 ECMO patients with 51 matched controls, wherein peak meropenem concentrations were drawn at 2 h after infusion and immediately prior to a subsequent dose, found no differences in distribution volume, half-life, or clearance [73]. Another study comparing 11 ECMO patients to historical controls sampled meropenem at 15, 30, 45, 60, 120, 360 and 480 min, finding a slight decrease in clearance and increase in volume of distribution [110]. Recommended dosing in this population involves a 1 g load followed by 1 g q8h [43], or 2 g q8h [110]. Higher doses of meropenem may be employed, and a regimen totaling 6g/d showed to be slightly superior in achieving MIC to standard dosing. Continuous infusion of 3–6/g has been recommended in patients with increased clearance or resistant organisms [110]. Notably, 6.1% of patients did not achieve target MIC compared to 0% of those receiving a higher-dose regimen [80]. High dosage may be considered in patients necessitating higher MICs [111]. Notably, in one study involving patients undergoing renal replacement therapy, MIC levels < 1 were associated with increased mortality [112].

Imipenem, also within the carbapenem classification, has also been studied. One study including 247 ECMO patients found lower plasma levels and higher dosing recommendable [111]. Others trialed 0.5 g every 6 h in 10 ECMO vs. 18 non-ECMO patients, sampling after the fourth dose and finding an increased distribution volume yet decreased clearance, which also recommended increased dosing [112]. Overall, increased dosing may be required, up to 4 g/day in reported cases [43,103], with prolonged infusion of 1 g over 4 h q6h as a recommendable strategy [112].

Cefazolin was reported as being sequestered in the ECMO circuit, although the physical properties of the circuit were critical [3,109]. Up to 84% of the cefazolin in vitro studies could be sequestered [3]. In the case series of ECMO patients, cefazolin clearance was significantly higher. The level of unbound cefazolin was higher and was most likely compensated by high Pk variability and changes in the volume of distribution [93]. In another case report, cefazolin pharmacokinetics was not changed [95]. These two studies may be reconciled, as Booke et al. demonstrated high interindividual variability in cefazolin kinetics [93]. In summary, adjusting cefazolin does not need to be performed in ECMO patients.

Ceftazidime demonstrated to be unaffected in serum dynamics in 30 ECMO patients compared to 75 non-ECMO ICU patients (from a mean age of ECMO 47.7 vs. 61.2 for non-ECMO in a prospective study). Consequently, adult dosing recommendations are to use a loading dose of 2 g intravenously and to adjust the dosing based on GFR (more than 30 = 6 g/24 h; less than 30 = 4 g/24 h) [80].

For teicoplanin, 89% of the drug can be sequestered, according to an in vitro study of the primed circuit [110]. Two in vivo studies agreed that the drug’s loading has to be increased to 12 mg/kg to achieve therapeutic concentrations [113,114].

Ciprofloxacin belongs to the fluoroquinolone family. These drugs are lipophilic bactericidal, exhibiting a volume of distribution mostly unaffected by critical illness [47] and low protein binding [43]. The half-life of fluoroquinolones may be decreased in critical illness, necessitating more frequent dosage [47]. Although lipophilic, ciprofloxacin has exhibited minimal circuit sequestration in studies [82]. A recommended loading dose of ciprofloxacin is 800 mg followed by 400–600 mg q8h [43].

Piperacillin should be used with caution in ECMO patients, wherein they tend not to achieve the desired therapeutic targets in these patients. One single-center study showcased this phenomenon, wherein piperacillin–tazobactam-treated patients were less likely to achieve a prespecified ×4 MIC (48% vs. 13% in non-ECMO patients) [75]. A loading dose of 4.5 g is recommended, followed by 4.5 g q6h or doses as per clearance [43].

Linezolid patients receiving linezolid may also show a tendency to not achieve desired plasma levels (35% vs. 15%) [80]. Nevertheless, if selected, a linezolid loading dose of 600 mg followed by 1800 mg/d continuous infusion is recommended [111]. 

Caspofungin falls under the echinocandin classification as a lipophilic antifungal. However, reports regarding circuit sequestration are conflicting. For example, circuit loss secondary to sequestration may be as high as 43%, while others have deemed this drug as unaffected by ECMO [44,82]. One prospective observational study in post-transplant patients compared 12 ECMO patients to seven matched controls. Sampling was performed after the second and third caspofungin dose, finding no significant pharmacokinetics [114]. Hence, the usual dosing of 70 mg loading with subsequent 50 mg/d dosing may be sufficient [111]. Prior studies have noted a Vd for caspofungin within normal parameters in these patients [81].

Micafungin, another echinocandin, exhibited similar results, with sequestration gauged around 45–99% [110]. However, in one observational study on 12 ECMO patients, micafungin sampling at 1, 3, 5, 8, 18 and 24 h after infusion yielded no differences in clearance or distribution volume [115]. No consensus on dosing recommendations for micafungin were available at the time of writing [111].

Voriconazole is a triazole antifungal commonly employed in *Aspergillus* sp. infection. While it was previously assumed that high circuit losses could be expected due to the drug being highly lipophilic, one large retrospective study found no significant pharmacokinetic changes during ECMO. The in vitro study showed significant absorption by circuit [94,110]. Some demonstrated sequestration with up to 71% circuit losses, with later saturation and redistribution reported [82]. The median dose was 9.2 mg/kg; however, higher dosing might be necessary, given that a total of 56% of patients in this study did not reach target levels compared to 39% of the control group [102].

In addition, a member of the triazole family, fluconazole, has exhibited minimal sequestration. However, data sufficient for dosing recommendations remain lacking [111].

## 7. Interaction of Antibiotics with Other Treatments 

Standard precautions regarding drug interactions apply, as patients on ECMO are bound to receive diverse agents during their course. More importantly, nearly 50% of all ECMO patients may necessitate renal replacement therapy (RRT) during their illness, further confounding antibiotic dosing [80,83]. The renal replacement circuit may be spliced into that of the ECMO, foregoing the need for further cannulation, although various access strategies have been employed [83]. Similar to ECMO, RRT mediates pharmacodynamic changes that must be accounted for when dosing strategies are selected. These alterations may be secondary to both drug properties or may be inherent to the RRT circuit itself.

Further compounding this issue in patients receiving concurrent ECMO and RRT, commonly utilized formulae employed for dosing calculations such as EGFR and Cockcroft-Gault may lose accuracy in this setting [19]. In addition, subtherapeutic levels may be observed in up to 25% of patients undergoing RRT alone [112]. This further highlights the exquisite need for therapeutic drug monitoring as necessary for management.

Various drug properties, including molecular size, protein binding, distribution volume, and metabolism, affect dialysis-mediated removal. In general, highly protein-bound drugs possess large molecular weight or volume of distribution, and/or non-renally cleared medications are least likely to be impacted by RRT [19]. Both the schedule or duration of RRT and effects exerted by the RRT circuit itself, including the use of high flow filters, may affect RRT-mediated clearance [19]. A rising estimated total renal clearance (eTRCL) correlated with lower trough concentrations for all antibiotics in one recent study [112]. 

While there is a paucity of data regarding pharmacodynamics in patients receiving concurrent ECMO and RRT, it is safe to suggest that the importance of therapeutic monitoring is further enhanced in these patients. A sieving coefficient can be determined for a drug if both plasma and ultrafiltrate concentrations are known (ultrafiltrate/plasma) [83]. This could be a potential avenue for further determining the interplay between ECMO, RRT, and antibiotic levels in the future. 

Impala and other devices are not present in the data pertinent to the concomitant application of ECMO and cytokine absorption technique. 

## 8. Effectiveness of Antibiotic Therapy in ECMO Patients

Several reports demonstrated that ECMO did not interfere with successful treatment of bacterial infections. However, given that these are mostly case reports, there is a lack of randomized controlled studies comparing success rates between patients treated with a similar regimen of antibiotics on ECMO vs. patients treated with mechanical ventilation. The CEASAR study was the only one designed in a way that demonstrated the superiority of transferring the patient to the specialized center vs. regional care [4]. There was no significant difference between the mechanical ventilation arm and ECMO once patients were transported to the reference center. Another study demonstrated a lack of mortality as well [115]. Although this study was followed by metanalysis incorporating large case reports, CEASAR may suggest that antibiotic therapy may be equally effective while the patient is on ECMO [1]. This is somewhat puzzling considering the large body of evidence suggesting sequestration of the antibiotic’s changes in Vd and Pk among many antibiotics. However, in at least one study, free antibiotics were significantly higher, offsetting the lower overall levels [93]. Another hypothesis is that bactericidal antibiotics are high enough to provide a bacteriostatic level while the immune system can clear the pathogen. 

The definite answer may be difficult to study as comparing study design in the CEASAR format may be unfeasible due to the ethical constraint. However, it is also interesting that since the study’s conclusion, no similar study design was followed, while ECMO proponents relied on case reports.

## 9. Conclusions

Antibiotic therapy success may be difficult to achieve in the ECMO patient. The emergence of critical care illness creates a difficult condition at the baseline. The variability introduced by the circuit further complicates clinical decision making. Although we suggest utilizing good stewardship in antibiotic dosing combined with drug level monitoring, one must realize that these methods are likely to be insufficient to predict the appropriate regimen in the ECMO situation (Figure 4). Utilizing the software targeting the drug therapy may not be helpful, as several variables seem to compensate for each other, in cases of cefazolin at least [93]. The monitoring of the clinical response may be optimal yet difficult, considering that ECMO may blunt some responses (fever) while unpredictably affecting others (procalcitonin levels).

## Figures and Tables

**Figure 1 antibiotics-11-00338-f001:**
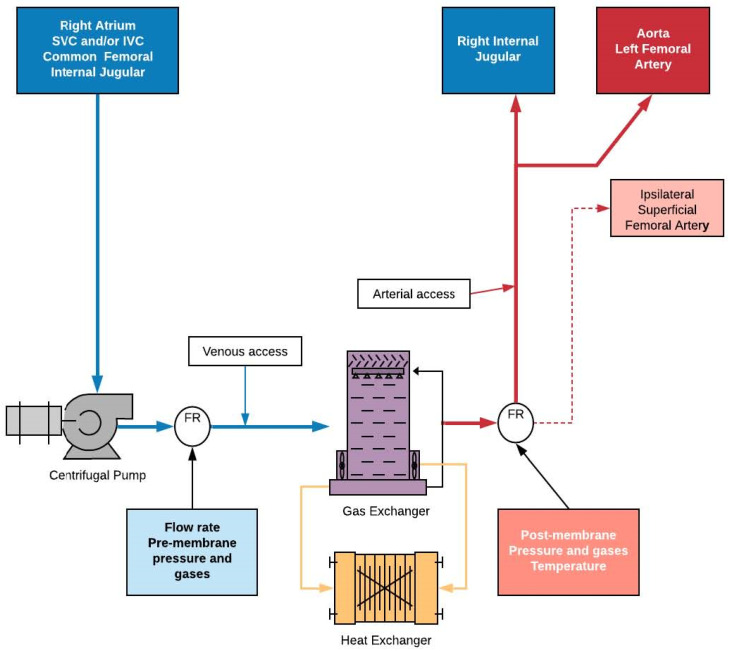
Sample VV ECMO circuit and possible cannulation sites.

**Figure 2 antibiotics-11-00338-f002:**
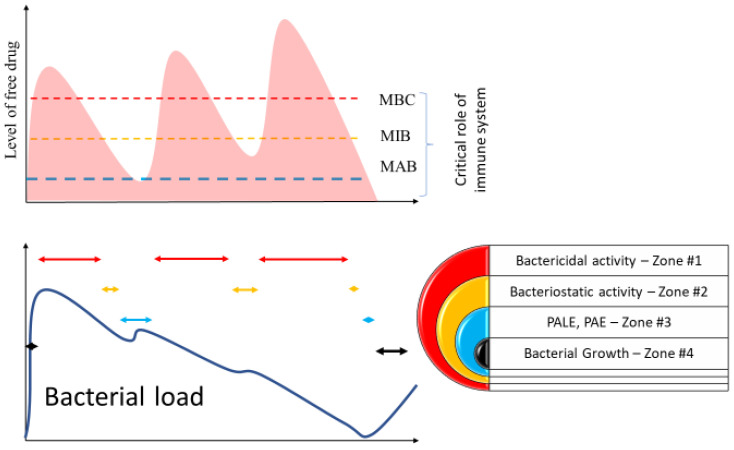
The level of the antibiotics are gradually increasing over the time to cross the MAB, MIB and MBC threshold, but only above MBC thresholds can the antibiotics eradicate infection instead of augmenting the immune system function.

**Figure 3 antibiotics-11-00338-f003:**
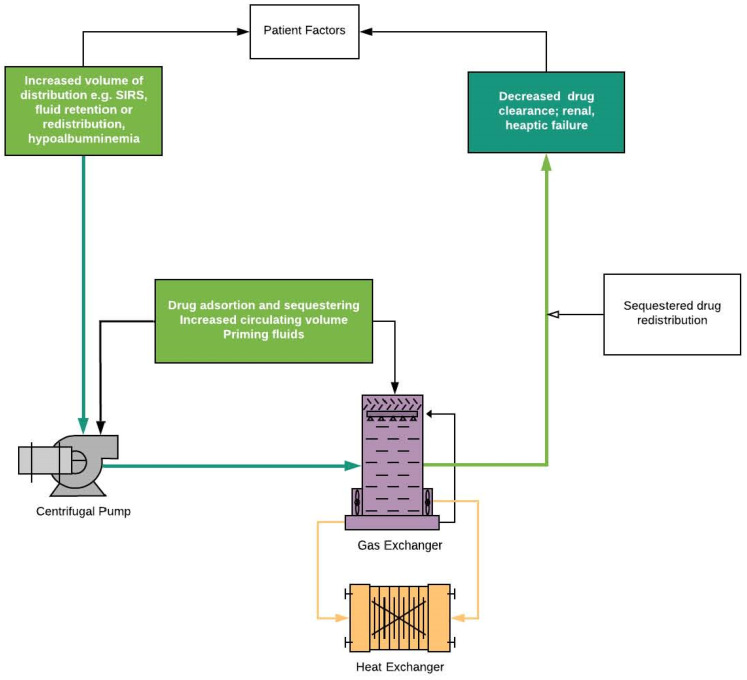
ECMO-specific factor affecting drug distribution.

**Figure 4 antibiotics-11-00338-f004:**
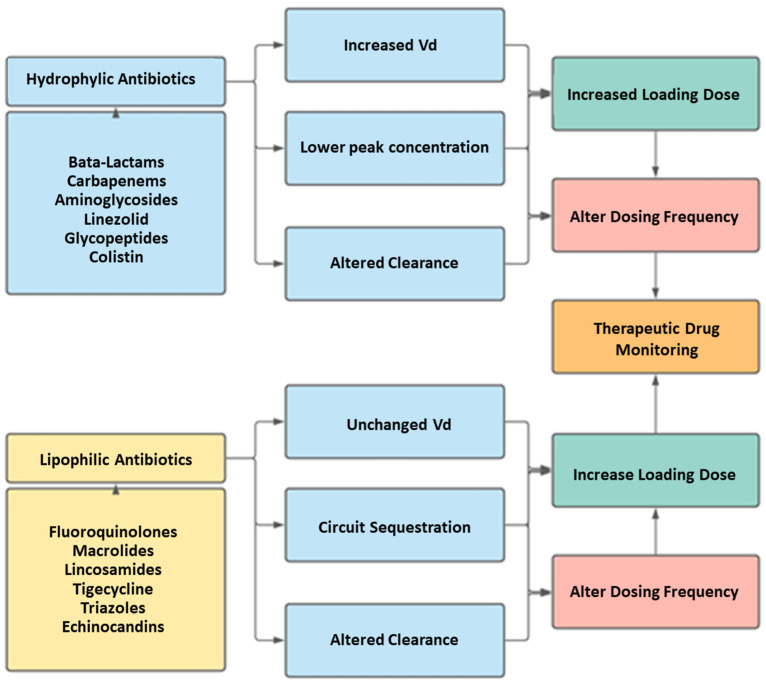
General recommendations regarding antibiotics as classified by hydrophilicity or lipophilicity [47,81,82,102].

**Table 1 antibiotics-11-00338-t001:** Selected antibiotics are divided into hydrophilic and lipophilic.

Hydrophilic	Lipophilic
Aminoglycosides	Fluoroquinolones *
β-lactams	Clindamycin
Glycopeptides	Tigecycline
Linezolid	Caspofungin
Colistin	Voriconazole

* Note that despite fluoroquinolones being described as lipophilic, the circuit loss rate for ciprofloxacin has been described as negligible. Thus, lipophilicity is not the only predictive factor for circuit sequestration [47,82].

## Data Availability

Not applicable.

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
