# Peer review of "Antibiotics and ECMO in the Adult Population—Persistent Challenges and Practical Guides"

_antibiotics, 2022, doi:10.3390/antibiotics11030338_

Round 1

Reviewer 1 Report

Dear Authors:

In this manuscript by Gomez has summarized pharmacokinetics and pharmacodynamics of antibioticsduring ECMO, emphasizing practical application and review of patient-, illness-, and ECMO hardware-related factors, which is excellent organized and will guide doctors to improve the usage and regimen of antibiotics during ECMO treatment in the future. Strongly suggest for publishing.

Best,

Author Response

Reviewer 1 In this manuscript by Gomez has summarized pharmacokinetics and pharmacodynamics of antibioticsduring ECMO, emphasizing practical application and review of patient-, illness-, and ECMO hardware-related factors, which is excellent organized and will guide doctors to improve the usage and regimen of antibiotics during ECMO treatment in the future. Strongly suggest for publishing. Much appreciate reviewer 1’s comments,

Reviewer 2 Report

Good work. nobody revision is necessary.

Author Response

Good work. nobody revision is necessary.

Much appreciate reviewer 2’s comments

Reviewer 3 Report

This manuscript is written well, so I recommend to publish this after revision.

1) The purpose of this REVIEW is not clear. Please show me more details. Why do you need this REVIEW?

2) Authors showed these words in Table 1.

[Selected antibiotics are dichotomized into hydrophilic and lipophilic.] 

However, is this need to show here? If so, could you show this more specific? I could not understand this here table well. 

Author Response

1) The purpose of this REVIEW is not clear. Please show me more details. Why do you need this REVIEW?

2) Authors showed these words in Table 1.

[Selected antibiotics are dichotomized into hydrophilic and lipophilic.] 

However, is this need to show here? If so, could you show this more specific? I could not understand this here table well. 

Reply

1. The purpose of this review is to evaluate the interactions between ECMO and antibiotic therapy. “. However, several covariables emerge during ECMO implementation, potentially jeopardizing the success of antimicrobial therapy” We further emphasize the importance of understanding said interactions given the frequency of infection encountered during or as the root cause of necessity for ECMO implementation “A common indication for ECMO is acute respiratory distress syndrome (ARDS), most commonly from infectious etiopathogenesis[5, 7, 8]. In addition, sepsis is considered the indication for ECMO deployment in some cases[10, 13]. Alternatively, patients undergoing ECMO may develop infectious complications by-products of implementations[6, 7, 14]. The risk is relatively elevated considering the presence of invasive cannulation and emergence of immunosuppression secondary to critical care illness, and the introduction of mechanical support devices[15-17]. “

  1. Given any specific antibiotic’s lipophilicity or hydrophilicity dictates in large part it’s pharmacokinetic properties, a brief table presenting which fall under each category is called for in a discussion regarding pharmacokinetics. We have changed the word “dichotomized” to “divided into” in order to avoid confusion.